# Predicting the effects of reservoir water level management on the reproductive output of a riparian songbird

**Mathew Hepp[1], Eirikur Palsson[2], Sarah K. Thomsen[1¤], David J. Green[1]***

**1** Department of Biological Sciences, Centre for Wildlife Ecology, Simon Fraser University, Burnaby, BC, Canada, **2** Department of Biological Sciences, Simon Fraser University, Burnaby, BC, Canada

¤ Current address: Department of Integrative Biology, Oregon State University, Corvallis, OR, United States of America

* djgreen@sfu.ca

**Data Availability Statement:** The data for the model is either provided within the manuscript or can be publically downloaded (https://wateroffice. ec.gc.ca; station 08NE104). The code for the individual based-model is provided as one of the

## Abstract

Dams and reservoirs alter natural water flow regimes with adverse effects on natural ecosystems. Quantifying and reducing these effects are important as global demands for energy and water, and the number of dams and reservoir, increase. However, costs and logistic constraints typically preclude experimental assessment of reservoir effects on the environment. We developed a stochastic individual-based model (IBM), parameterized using empirical data, to estimate the annual productivity of yellow warblers that breed in riparian habitat within the footprint of the Arrow Lakes Reservoir in British Columbia, Canada. The IBM incorporated information on breeding phenology, nest site selection, brood parasitism, daily nest survival, re-nesting probabilities and post-fledging survival. We used the IBM to estimate the effect of four different water management scenarios on annual productivity. We found that the IBM accurately estimated average nest success (0.39 ± 0.10 SD), the proportion of females that produced at least one fledgling during a breeding season (0.56 ± 0.11), and annual fledging success (2.06 ± 0.43) under current conditions. The IBM estimated that reservoir operations currently reduce the annual productivity of this population by 37%, from an average of 1.62 to 1.06 independent young/female. Delaying when reservoir water levels reach 435m asl (the minimum elevation occupied by yellow warblers) by approximately 2 weeks was predicted to increase annual productivity to 1.44 independent young/female. The standardized effect on annual productivity of reducing the maximum elevation of the reservoir so that yellow warbler habitat is not inundated (Cohen's d = 1.52) or delaying when water is stored (Cohen's d = 0.83) was primarily driven by inundation effects on post-fledging survival. Reservoir operation effects on breeding birds will be species specific, but this IBM can easily be modified to allow the environmental impacts on the entire breeding bird community to be incorporated into water management decisions.

Supplementary Information files. The basic breeding data has been deposited with Dryad https://doi.org/10.5061/dryad.rfj6q5792.

**Funding:** Funding for this work came from National Science and Research Council Discovery grants to DJG (RGPIN 2009-261899, 2014-05798, 2019-05513) and EP (RGPIN-2016-04625) and a Columbia Basic Fish and Wildlife Compensation Program (http://fwcp.ca) grant to DJG (SPI 4639). Fieldwork was supported by an Environment and Climate Change grant and contribution to the Centre for Wildlife Ecology, Simon Fraser University (GCXE19SO059). Fieldwork was also indirectly supported through a BC Hydro Water License Requirements contract to Cooper Beauchesne and Associates Ltd (CLBMON-36). The funders had no role in study design, data collection and analysis, decision to publish, or preparation of the manuscript.

**Competing interests:** Cooper Beauchesne and Associates Ltd provided field accommodation and a food allowance for a graduate student and field assistant that collected data used in this paper in each year from 2008-2017. This does not alter our adherence to PLOS ONE policies on sharing data and materials.

## Introduction

Rivers and river systems are becoming increasingly modified by human activities [1, 2]. Dams created for hydroelectric power generation, flood control and domestic water use have modified over 50% of the world's rivers [3]. In North America alone, dams have been used to create more than 2000 reservoirs with a total storage capacity of over 1500 km³ [4]. The loss of riparian habitat and alteration of water flow and flooding dynamics associated with reservoirs and dams [5] has had a large impact on many forms of wildlife [6, 7].

A number of avian species are dependent on riparian habitat during the breeding season and habitat loss has been argued to be a driver of population declines for many of these species [8–11]. Where riparian habitats persist in reservoir basins, the operations of the reservoir may have negative effects on the productivity of birds that breed there [12, 13]. For example, seasonal inundation of riparian habitat may reduce the amount of time available for reproduction and flood nests before eggs hatch or nestlings are able to fly [14]. However, quantifying the effects of reservoir operations on the annual productivity of riparian birds is challenging. Studies that monitor nests can document flooding (e.g. [15]), but need to evaluate the extent to which flooding reduces nest success that may also fail for other reasons [16], allow for the possibility that individuals of multi-brooded species can re-nest, and incorporate reservoir operation effects on the survival of fledglings [17]. Long-term monitoring studies of species of concern can sometimes document changes in annual productivity associated with variation in the management of reservoir water levels (e.g. [13]), although annual variation in brood parasitism or nest predation rates could potentially mask any effects on annual productivity. Meanwhile, financial costs and logistic constrains typically preclude experimental assessment of reservoir operation effects on riparian bird populations.

Individual-based models [18, 19] that simulate the reproduction of females across an entire breeding season are a tool that can be used to predict how management scenarios influence the demography of populations. For example, Ratcliffe et al. [20] used an individual-based model to examine how the management of water levels in the Ouse and Nene Washes, that are occasionally flooded to prevent the inundation of surrounding farmland, influences the productivity of black-tailed godwits *Limosa limosa* in eastern England. Their model showed that flooding of wet meadows reduced productivity by forcing godwits to nest in nearby arable fields where daily nest and chick survival rates were low. The flood-dependent productivity estimates produced by their model were sufficient to explain the contrasting population trends observed at the two locations. More recently, Buckingham et al. [21] used an individual-based model to examine whether manipulating when and how grass fodder crops are harvested would boost the reproductive output of Eurasian skylarks *Alauda arvensis* in southwestern England. Their model predicted that delaying mowing and raising cutting heights would increase reproductive output, but that females breeding on silage fields would still produce insufficient independent young to replace the annual loss of adults. These individual-based models are obviously data-intensive in that they require information on breeding phenology and the estimation of a number of breeding parameters (e.g. clutch size, daily nest survival rates, incubation and nesting periods, re-nesting rates, and post-fledging survival) that may vary seasonally, with habitat type or environmental conditions. However, these breeding parameters may be obtained from the literature or estimated using data collected over a relatively short period (e.g. [21, 22]). The models can also be validated by showing that they generate realistic biological outcomes and sensitivity analyses can be used to explore the sensitivity of model results to uncertainty in parameter estimates [19]. Individual-based models therefore provide an effective and cost-efficient method to evaluate a wide range of management actions without the need for prohibitively expensive field trials.

In this study we develop a stochastic, spatially implicit individual-based model to estimate the annual productivity of female yellow warblers *Setophaga petechia* breeding in riparian habitat within the drawdown zone of the Arrow Lakes Reservoir in British Columbia, Canada. The model incorporates annual variation in nest initiation dates, seasonal variation in clutch size and re-nesting probabilities, the effects of brood parasitism, and measurements of daily nest survival rates and post-fledging survival to independence. We parameterize and validate the model using an empirical dataset from a long-term study of this population that spans 12 years (2005–2017). We use the model to predict how changes in the management of water levels in the Arrow Lakes Reservoir would influence variation in annual productivity and assess the sensitivity of the model predictions to uncertainty in our observed breeding parameters.

## Methods

This research was approved by the Simon Fraser University Animal Care Committee (1200B) and conducted under an Environment and Climate Change Canada scientific permit (10759H).

## Study site and species

We have studied yellow warblers in riparian habitat along the Arrow Lakes Reservoir, in the Columbia River Valley near Revelstoke, British Columbia, Canada (50°58'56"N 118°20'00"W) since 2004. This reservoir system is bounded by the Monashee and Selkirk mountain ranges. Water levels in the reservoir fluctuate between a low water elevation of 419.65m and a maximum of 440.1m (i.e. full pool) and are controlled by operations (both for hydroelectric and flood control purposes) at Hugh Keenleyside Dam downstream and Revelstoke Dam upstream of the reservoir. Water levels typically rise in May and June and peak in July, but the timing of the rise and the maximum water level in the reservoir varies across years [16]. Historical daily water level data for Arrow Lakes Reservoir (station 08NE104) can be downloaded from https://wateroffice.ec.gc.ca. We established three plots, of 24–30 hectares (435m-441.7m in elevation), in and alongside the drawdown zone of the reservoir. Vegetation at these sites is dominated by black cottonwood tree stands at high elevations and willow species at lower elevations. This habitat, even though partially in the drawdown zone of the reservoir, closely resembles what is found in natural riparian habitat. A more detailed description the study sites and vegetation can be found in [15].

The yellow warbler is a medium-sized wood warbler that winters in Mexico and South America and breeds across most of North America from the low arctic south to Mexico. Migrants are first seen in California during April [23], and the first birds arrive on our study sites in early to mid-May [24]. Males arrive a few days before females and prefer to settle in territories with more riparian shrub and tree cover and greater shrub diversity [15]. Females build nests in a variety of shrubs (predominantly Salix spp.) and lay clutches that typically contain 4–5 eggs (range = 2–6). Brown-headed cowbirds *Molothrus ater* parasitize, on average 19% of nests that are initiated each year (range = 3–33%), typically laying a single egg after removing one host egg from the nest. Female yellow warblers subsequently abandon 19% of parasitized nests. Brood parasitism does not influence nest success but reduces the number of young that fledge from successful nests by, on average, 1.3 offspring [25]. Females that lay 4 eggs have a nesting cycle that lasts approximately 23 days; eggs are laid daily (1 egg = day 1), incubation commences on the penultimate egg (= day 3), nestlings hatch after an incubation period of 11 days (= day 14) and fledge after a nestling period of 9 days (= day 23) [26]. Females can nest multiple times during a breeding season (maximum = 3) and may renest

after both failed and successful nesting attempts. Fledglings are independent of their parents after approximately 21 days [17].

## Breeding biology, daily nest survival rates and re-nesting probability

We compiled data on the breeding biology of yellow warblers collected over twelve years (2005–2006 and 2008–2017). In these years we monitored all breeding pairs at the three study sites from early May, when males first started establishing territories, to late July, when the last nest fledged or failed. We typically caught and banded birds that established territories soon after they arrived using 6m to 12m mist nets and playback of male song and female chipping calls. We also caught incubating and provisioning females in mist nets placed close to their nest. We banded birds with a unique combination of color bands and a Canadian Wildlife Service-issued aluminum band for identification during the breeding season and in subsequent years.

We attempted to find and monitor all nests built by every female during the breeding season. We recorded the nest height and location of each nest using a GPS unit (Garmin GPSMAP 76CSx). We subsequently monitored nests every 1–4 days to determine when the first egg was laid, the clutch size, whether a nest was parasitized by a Brown-headed cowbird, the hatch date, the date when the nest failed or when nestlings fledged. If nests were found during incubation or the nestling stage, we estimated the first egg date using the hatch date or nestling age assuming that eggs were laid daily and incubated for 11 days. We noted whether nests were abandoned or failed due to predation, flooding or other causes. We assigned a date of failure as the midpoint between nest checks. Where possible we banded nestlings seven days post-hatch and assumed that the number of nestlings present on this date was the number that fledged if there was no evidence of predation and parents were observed feeding or defending fledged young.

We estimated daily nest survival rates for all nests monitored over the 12 years of this study, excluding only nests that were parasitized by Brown-headed cowbirds or that failed due to flooding, using Program Mark [27]. We constructed models (n = 5) to evaluate whether daily nest survival was constant, varied across the nesting season as a linear or quadratic function of the day of the season, varied with nest age, or differed during periods when the nest contained eggs (laying/incubation) or nestlings. Models were ranked based on their AICc and compared using AIC weights.

We evaluated whether the ground elevation at the nest site, nest height, clutch size and re-nesting probability (after nests failed or fledged young) varied seasonally using generalized linear models implemented in R [28]. Models were fitted using either a gaussian distribution (ground elevation and nest height), a poisson distribution and a log link (clutch size), or a binomial distribution and a logit link. Models examining variation in clutch size excluded data if the nest was parasitized by a Brown-headed cowbird, whereas models examining re-nesting probability included all nests that failed or successfully fledged young. We did not use mixed models that included female identity as a random term as the majority of females contributed only one line of data to each dataset (Clutch Size dataset—87% of females, Re-nest after fail dataset—51% of females, Re-nest after success dataset—97% of females).

## The individual-based re-nesting model

We created a stochastic, spatially implicit individual-based model to predict the annual productivity of female yellow warblers breeding in riparian habitat in and along the drawdown zone of the Upper Arrow Lake reservoir (Fig 1). The code for this model, written in MatLab

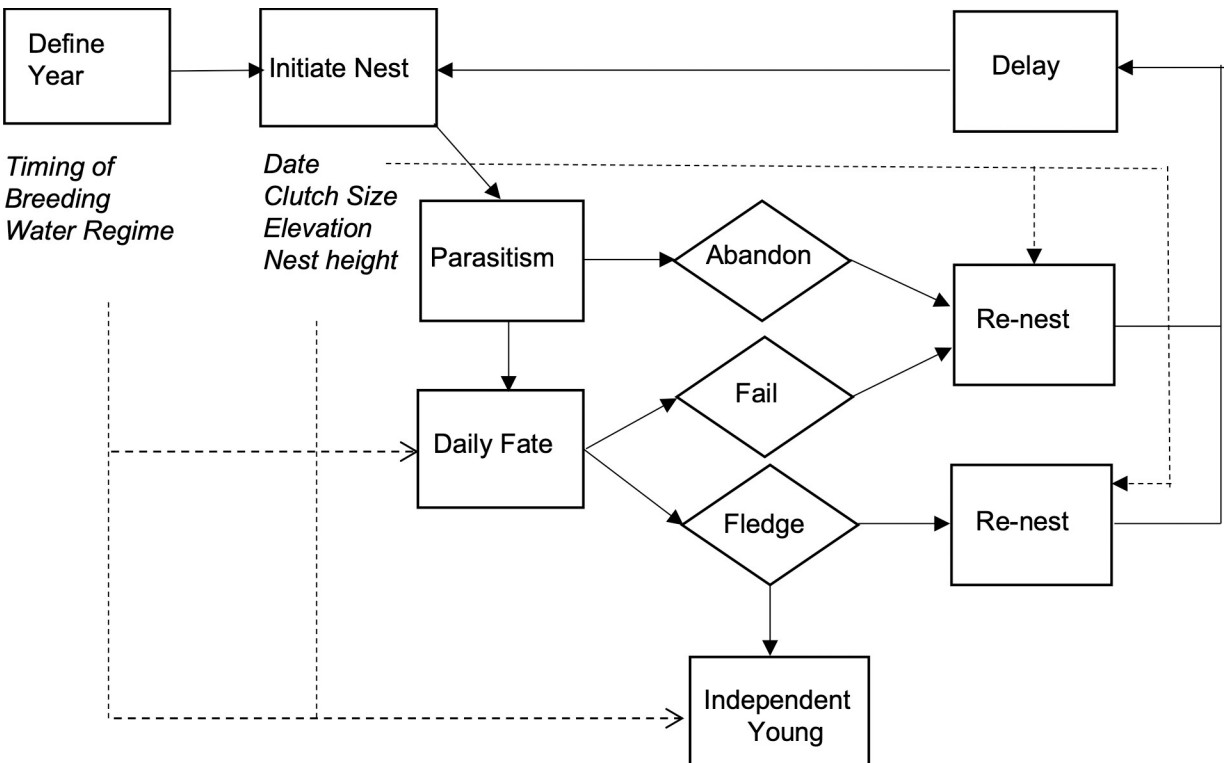

**Fig 1. Schematic of the individual-based model of yellow warbler productivity.** Boxes are model processes, diamonds are nest fates, and italics are parameters or input data. Solid arrows indicate model flow path, and dashed arrows indicate variable effects. The model is initialized by defining the year and, for each female, the nest site ground elevations and nest height. Nests can be parasitized and abandoned and have a daily fate that is dependent on the elevation of the nest, reservoir water level and daily nest survival rates. Females may re-nest after the failure or success of a previous attempt.

[29], is available in the S1 Data. We parameterized the model using data from this population collected between 2005 and 2017 (see Results, Table 1). Our model simulated the breeding of 35 females per year (the average number breeding at our three study sites) for 50 years. For each year the model run is initialized by selecting (i) a year from a 50-year dataset that describes the daily reservoir water level during the breeding season (see below), and (ii) the timing of breeding in a given year. The average date female yellow warblers initiate their first clutch varies with conditions on spring migration [30]. For each female their breeding season is initialized by selecting (i) the date that they lay their first egg of the season (relative to other females in that year), (ii) the ground elevation of the nest site, and (iii) the height of the nest. Date, ground elevation and nest height are randomly chosen from a normal distribution, where the mean and SD are calculated from the monitoring data. Female then lay one egg per day until a clutch is complete with clutch size being a function of the date that the first egg is laid. Nests may be parasitized during egg-laying (on the day before the last egg is laid), which may prompt females to abandon the nest, both with a fixed probability. Brood parasitism is assumed to reduce the size of the clutch by 1 egg. For each day after a nest is initiated a nest may fail due to flooding, a fate that is dependent on the reservoir water level, nest elevation (ground elevation + nest height), or fail due to other causes with a probability that is a function of the nesting stage (eggs vs nestlings). Nests that survive the complete nesting cycle fledge young, with the number fledging dependent on the combined probability of hatching failure and nestling death. Fledged young survive to independence at 21 days with a probability that is

**Table 1. Parameters used in the yellow warbler individual-based model.**

| Parameters | Stochastic Process | Value/equation | Notes | Data source |
|---|---|---|---|---|
| **Initialization** | | | | |
| Reservoir water levels | Uniform; f(Year) | | n = 50 years | BC Hydro |
| Breeding phenology | Normal | DOY[a] = 158 ± 2.3 days (range 155–163) | varies with migration conditions | this study, |
| Mean date first egg laid | | | | Drake et al. 2013 |
| Female date of first egg | Normal | 0 ± 7 days range (-19-+26) | | this study |
| Nest site ground elevation | Normal | 438.5±0.9 range (435.6–441.7) | | this study |
| Nest height | Normal | 2.1 ± 1.1 m range (0.3–6.0) | excluding rare (16 of 522) nests in cottonwoods | this study |
| **Nest cycle** | | | | |
| Clutch Size (CS) | Fixed; f(DOY) | CS = 9.60–0.033*(DOY) | | this study |
| Days to fledge | Fixed; f(CS) | Days = CS+18 | n = 39 nests | Martin et al. 2019 |
| Parasitism probability | Bernoulli | 0.19 | n = 218 nests | Rock et al. 2013 |
| Abandon probability | Bernoulli | 0.19 | n = 42 nests | Rock et al. 2013 |
| Daily nest survival | Bernoulli | | | this study |
| • **egg** | | 0.9712 | | |
| • **nestlings** | | 0.9577 | | |
| Survival to independence | Bernoulli | | | |
| • **nest site inundated** | | 0.214 | n = 26 young | Hepp et al. 2018 |
| • **nest site dry** | | 0.729 | | |
| **Re-nesting** | | | | |
| Re-nest after failure | Bernoulli; f(DOY) | Logit(p) = 35.534–0.208*(DOY) | | this study |
| Delay after failure | Fixed | 6 days | range (3–14) | this study |
| Re-nest after success | Bernoulli; f(DOY) | Logit(p) = 34.324–0.211*(DOY) | | this study |
| Delay after success | Fixed | 7 days | range (5–9) | this study |

[a] DOY = Day of year, where January 1 = 1

dependent on whether the reservoir water level is above or below the elevation at the nest site. Females that abandon a nest, have a nest that fails or successfully fledge young may re-nest; the probability of re-nesting is a function of the date that a nest is abandoned/fails or the date young fledge. However, females may only initiate three nesting attempts in a year. Females that do re-nest initiate a new nesting attempt after a delay that varies in duration depending on whether the previous nest failed or was successful. Females are then assigned a new nest site (ground elevation) and nest height and start a new nesting attempt. Females that do not re-nest, after the first or second nesting attempt are recorded as having completed their breeding season.

## Water management scenarios

We used the individual-based re-nesting model to predict annual productivity under four scenarios that differ in when and whether reservoir water levels in the Arrow Lakes Reservoir reach an elevation of 435m a.s.l. (the lowest elevation at which willow shrubs are found and yellow warblers nest in the drawdown zone).

### Scenario 1. Long-term conditions

Reservoir water levels reflect water levels observed in the reservoir over the last 50 years. BC Hydro provided daily water levels in the Arrow Lakes Reservoir from April 1, 1968 to December 31, 2017. Over this time period, water levels did not reach 435m a.s.l in 7 of 50 breeding

seasons. In the remaining years, the median date when reservoir water levels reached 435m a.s. l. was June 12 (DOY = 163). Under this scenario years in the model run are selected at random, with replacement, from the entire dataset.

### Scenario 2. Early fill years

Reservoir water levels rise relatively early in the breeding season. We defined early fill years as years between 1968 and 2017 when water levels reach 435m a.s.l. by June 12. Under this scenario years are selected at random, with replacement, from the 22 years in the dataset that meet this criterion. This sample includes all of the last 10 years (2008–2017) and 11 of the 12 years empirical data was collected. In this subset of the data the median date when water levels reached 435m a.s.l was June 7 (DOY = 158).

### Scenario 3. Late fill years

Reservoir water levels rise relatively late in the breeding season. We defined late fill years as years between 1968 and 2017 where water levels do not reach 435m a.s.l. until after June 12 or never reach this elevation. This scenario reflects a management strategy designed to extend the period when flooding does not negatively impact shrub-nesting birds. Under this scenario years are selected at random, with replacement, from the 28 years in the dataset that meet these criteria. This subset of the data includes the 7 years where water levels do not reach 435m a.s.l, and in the remaining years the median date water levels reached 435m a.s.l was June 21 (DOY = 172).

### Scenario 4. Zero flooding

This scenario estimates annual productivity in the absence of reservoir effects on female yellow warbler breeding success. Reservoir water levels in zero-flooding years never exceed 435m a.s. l.

## Model validation, predicted effects and sensitivity

We evaluated our model by comparing the predicted annual nest success (proportion of nests that fledged young), breeding success (proportion of females that fledged $> 0$ young) and fledging success (average number of young fledged/female) of females under Scenario 2 with those observed between 2005 and 2017. We used the model output from Scenario 2 because 11 of the 12 years of observational data were collected in "early fill years". We were unable to compare predicted and observed productivity (average number independent young produced/ female/year) because we only used radiotelemetry to quantify the post-fledging survival of a subset of young in three years (see [17] for details of the radiotelemetry study).

We compare the predicted productivity of yellow warblers (the average number of fledglings produced per female and the average number of independent young produced per female) associated with reservoir water level management over the last 50 years (Scenario 1) or during the course of this study (Scenario 2) with alternatives that extend the period when flooding does not negatively impact yellow warblers (Scenario 3) or remove reservoir effects on yellow warblers altogether (Scenario 4). We report changes in the average number of fledglings and number of independent young produced per female alongside the standardized effects (Cohen's d) associated with changing from water management scenarios 1 and 2 to 3 and 4.

We assessed the sensitivity of our conclusions regarding reservoir impacts on annual productivity by comparing the effect of changing how reservoir water levels are managed if (i)

parasitism and nest abandonment rates were reduced or increased by 50%, (ii) daily nest survival rates were increased or decreased by 1%, and (iii) the post-fledging survival of young from nests inundated by rising reservoir water levels is doubled or made equal to the post-fledging survival of young from nests that are not inundated (i.e dry in Table 1). We selected 50% reductions/increases in parasitism rates and nest abandonment as this captures the observed inter-annual variation in these parameters, and 1% variation in daily nest survival to reflect biologically plausible variation in daily nest survival rates. We increased post-fledging survival at inundated nests to those that were not inundated to explore the importance of reservoir impacts at this stage of the breeding cycle.

## Results

### Breeding biology, daily nest survival rates and re-nesting probability

Each year, on average, 35 females (range 24–42) attempted to breed in our three study plots. Over the course of the study we monitored the breeding of 422 females and 522 nesting attempts (ca. 44 per year). We found nests located at ground elevations that ranged from 435.6 to 441.7m a.s.l. (mean ± sd = 438.5 ± 0.9). The ground elevation at the nest site was independent of date ($F_{1,521}$ = 0.71, p = 0.40). Nests were typically built in shrubs 0.3–6 m above the ground (mean ± sd = 2.1±1.1), although a few nests (16 of 522) were built high in cottonwoods found at higher elevations. Nest heights were independent of date ($F_{1,521}$ = 0.003, p = 0.95), and after excluding the 16 nests in cottonwoods, elevation ($F_{1,505}$ = 1.58, p = 0.21).

We found considerable variation in when females could initiate their first nesting attempt of the breeding season. Across years, the mean date that females initiated their first nesting attempt ranged from June 4 (DOY = 155) to June 12 DOY = 163 (mean ± sd = 158 ± 2.25). Within years, individual females initiated their first nesting attempt over a period that spanned approximately four weeks (mean ± sd, after controlling for year to year variation = 0 ± 7 days). Females laid clutches, initiated between 18 May (DOY = 138) and July 10 (DOY = 191), that contained 2–6 eggs (mean ± sd = 4.27 ± 0.67, n = 327), with smaller clutches being laid later in the season ($F_{1,326}$ = 94.8, p < 0.0001).

We found that 55% of all nests initiated failed to fledge young. Nests were rarely flooded as a result of rising water levels in the Arrow Lakes Reservoir (22 of 522 nests), occasionally abandoned (36 of 522 nests) and frequently depredated (234 of 522 nests). The proportion of nests that were successful varied from 0.39 in 2014 to 0.63 in 2005. Daily nest survival rates, estimated using data from 475 nests that were not flooded or abandoned as a result of nest parasitism by cowbirds, were higher during the laying and incubation period than the nestling period (Table 1). The nest survival model including a nest stage parameter (eggs or nestlings; AIC weight = 0.75) received 5x the support of a model that included a nest age term (AIC weight = 0.15), and 15x the support of a model where daily nest survival rates were independent of nest stage or age (AIC weight = 0.05). Daily nest survival did not vary seasonally; nest survival models that included a linear or quadratic date term received little support (AIC weights = 0.04 and 0.01, respectively).

Our study confirmed that female yellow warblers could re-nest after both failed and successful nesting attempts (129 cases after 287 failed attempts; 8 cases after 235 successful attempts), and therefore occasionally fledge two broods. However, the probability of re-nesting declined as the season progressed (Table 1; re-nest after failure, DOY effect, $\chi^2$ = 157.6, p < 0.0001; re-nest after success, DOY effect, $\chi^2$ = 12.9, p < 0.001). Females that initiated a new nesting attempt laid their first egg, on average, 6 days after a nest failed (range 3–14 days) and 7 days after young fledged (range 5–9 days).

Overall, we found that slightly more than half of the females (56%) monitored were able to fledge at least one young during a breeding season, and estimated that females fledged, on average 1.93 young per year. However, the percentage of females that were successful ranged from 43.8 in 2012 to 87.5% in 2005, and average annual productivity in these years ranged from 1.37 to 2.75 fledged young/female/year.

## Model validation

The individual-based model simulating the breeding of yellow warblers under reservoir conditions in Scenario 2 produced estimates of nest success, breeding success and fledging success that were close to those observed during the 12 years of empirical data collection. The model estimated that the mean proportion of nests initiated that fledged young was 0.39 ± 0.10 (95% CI across the 50 years = 0.21–0.55), the mean proportion of females that produced at least one fledgling over the course of the breeding season was 0.56 ± 0.11 (95% CI = 0.32–0.71), and that mean annual fledging success was 2.06 ± 0.43 young per female (95% CI = 1.28–2.72).

## Model estimated effects of water management on the breeding performance of yellow warblers

The individual-based model predicted that delaying the timing of when reservoir water levels reach 435m a.s.l by, on average, 2 weeks or avoiding the flooding of riparian habitat during the breeding season would have a small effect on fledging success and a large effect on the number of independent young produced by female yellow warblers breeding in riparian habitat in and along the drawdown zone of the Arrow Lakes Reservoir (Fig 2). The model predicted that changing from water management Scenario 2, that captures conditions over the last decade, to Scenario 3, where water levels do not reach 435m until after June 12, would increase the number of fledglings produced by 7% and the number of independent young produced by 36%. In contrast, changing from water management Scenario 2 to Scenario 4, where no riparian habitat is flooded, was predicted to increase the number of fledglings produced by 11% and the number of independent young produced by 58% (Fig 2). The standardized effect (Cohen's d) of these management actions on the number of independent young produced were estimated to be 0.83 and 1.52, respectively (Table 2).

## Model sensitivity

Conclusions regarding how changes to the water management regime influence the productivity of yellow warblers were insensitive to the rates of cowbird parasitism and nest abandonment. Increasing or decreasing parasitism and abandonment rates by 50% had surprising little effect on the average fledging success and annual productivity of female yellow warblers under any of the four water management scenarios (S1 Table). Decreasing and increasing daily nest survival rates by 1% had the expected effect on average fledging success and annual productivity but did not alter the differences in estimated productivity of yellow warblers in water management Scenario 1 or 2 and 3 or 4 (S1 Table). Water management scenario effects on productivity were, however, sensitive to the degree to which post-fledging survival was lower on territories that were inundated by rising water levels (flooded) than those that were not (S1 Table). Doubling post-fledging survival on inundated territories from 0.214 to 0.428 reduced the benefits of changing from water management Scenario 2 to Scenario 3 (revised Cohen's d = 0.67, 95% CI = 0.10, 1.2) or Scenario 4 (revised Cohen's d = 1.12, 95% CI = 0.52, 1.72). Removing inundation effects on post-fledging survival altogether further reduced the effect of changing from Scenario 2 to Scenario 3 (revised Cohen's d = 0.39, 95% CI = -0.17, 0.95) or Scenario 4 (revised Cohen's d = 0.55, 95% CI = -0.01, 1.12).

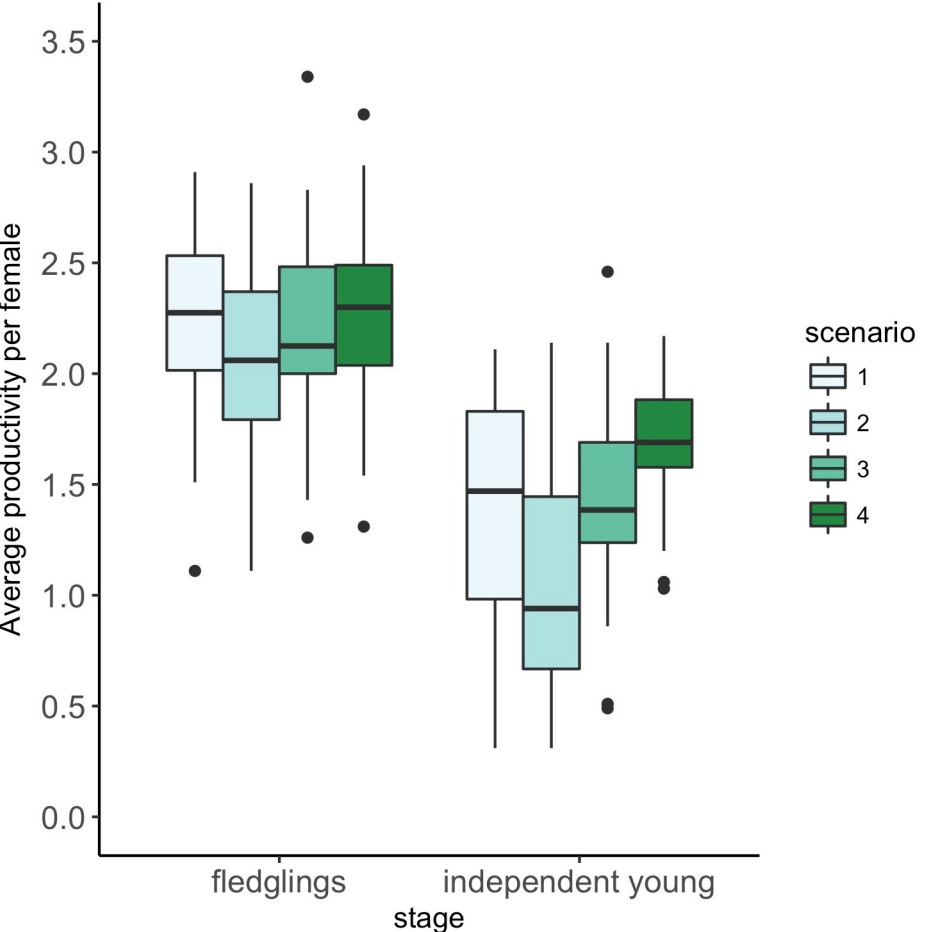

**Fig 2. Model estimated annual productivity of female yellow warblers under four reservoir water management scenarios.** Boxplots show estimates for 50 breeding seasons. Scenario 1 = long term conditions (ie the reservoir water levels observed between 1968 and 2017; Scenario 2 = early fill years (when water levels reach 435m a.s.l. by June 12); Scenario 3 = late fill years (when water levels do not reach 435m a.s.l. until after June 12); Scenario 4 = zero flooding (where reservoir water levels never exceed 435 m a.s.l). Boxes show the lower quartile, the median and the upper quartile. The whiskers extend to the smallest/largest value or 1.5 times the interquartile range, with the points showing outliers that extend beyond 1.5 times the interquartile range.

## Discussion

Individual-based models are a valuable tool that can be used to predict how populations of species of concern will be impacted by anthropogenic activity or respond to alternative management strategies [18]. We built an individual-based stochastic re-nesting model incorporating information on annual variation in breeding phenology, nest site selection, seasonal variation in clutch size, nest parasitism, predation rates, and re-nesting intervals to assess how fluctuations in reservoir water levels influences the annual productivity of a riparian dependent songbird. Our model provides estimates of breeding performance that are a good match to those estimated in the field and predicts that when (and whether) reservoir water levels reach the lower elevation of the willow habitat will have large effects on the numbers of young yellow warblers that survive to independence and recruit to the population. The extent to which reservoir water level management would benefit other riparian birds will depend on their breeding phenology, habitat selection decisions and life history. However, our model could easily be

**Table 2. Change in the estimated average productivity of female yellow warblers and the standardized effect sizes (Cohen's d, 95% CI) associated with changing reservoir operations from Scenario 1 or 2 to Scenario 3 and 4.**

| | New scenario | |
|---|---|---|
| **Initial scenario** | **Scenario 3 (Late fill)** | **Scenario 4 (Zero Flooding)** |
| Scenario 1 (Long-term conditions) | | |
| Change in average number of fledglings per female | 2.25 → 2.21 | 1.37 → 1.44 |
| Standardized effect size (d) | -0.10 | 0.08 |
| 95% CI | (-0.49, 0.29) | (-0.31, 0.47) |
| Change in average number of independent young per female | 2.25 → 2.28 | **1.37 → 1.68** |
| Standardized effect size (d) | 0.15 | **0.76** |
| 95% CI | (-0.40, 0.71) | **(0.19, 1.33)** |
| Scenario 2 (Early fill) | | |
| Change in average number of fledglings per female | 2.06 → 2.21 | **1.06 → 1.44** |
| Standardized effect size (d) | 0.37 | **0.54** |
| 95% CI | (-0.03, 0.76) | **(0.15, 0.95)** |
| Change in average number of independent young per female | **2.06 → 2.28** | 1.06 → 1.68 |
| Standardized effect size (d) | **0.83** | 1.52 |
| 95% CI | **(0.25, 1.41)** | (0.89, 2.15) |

Medium and large effect sizes are indicated in bold. Effect sizes of 0.5 would indicate that 69% of years under the new scenario are above the mean for Scenario 1/2, effect sizes of 0.8 would indicate that 79% of years under the new scenario are above the mean for Scenario1/2, and effect sizes of 1.5 would indicate that 93% of years under the new scenario are above the mean for Scenario 1/2.

adapted for other species, simplified where less data is available, and used to determine community level benefits of management actions.

Arrow Lakes Reservoir was created when the Hugh Keenleyside Dam was completed in 1968 and stores water primarily for enhanced flood control and downstream power generation. The presence of the dam and reservoir altered the hydrological regime as spring freshet water was impounded, increasing water levels throughout the year, with maximum elevations reached in July or August rather than June or July, followed by slower declines in the fall and winter until annual lows in the early spring. This seasonal pattern is repeated each year but is both more extreme and more variable than pre-1968 due to inter-annual variation in snowpack, spring melt timing, spring rainfall, and the demands for upstream and downstream power generation dictated by the Canadian Columbia River Treaty. Early-fill years are more common in the latter half of the time series (5 of 25 years from 1968–1992 compared to 20 of 25 years from 1993–2017); both operational decisions and the earlier onset of the snow melt and spring freshet and increased spring precipitation since the 1950's [31] may have contributed to this pattern. Management of water levels in Arrow Lakes Reservoir that delay when the reservoir fills by approximately 2 weeks (Scenario 3) would likely increase yellow warbler productivity to pre-1990 levels and partially mitigate climate change driven reductions in annual productivity of this yellow warbler population.

Reservoir water management impacts on the annual productivity of yellow warblers that breed within the footprint of Arrow Lakes Reservoir are driven by rising water level impacts on post-fledging survival. Flooding of nests is rare (ca 4%, this study) because clutches are initiated in advance of spring snowmelt, when water levels rise as water is stored in Arrow Lake reservoir [16]. Predation is therefore the major source of failure during the nesting period. Rising water levels do, however, coincide with the three-week post-fledging period when juveniles are dependent on their parents. Studies show this is a period of high mortality for songbirds

[32], and we have shown that the survival of radio-tagged fledglings and the recruitment of colour-banded fledglings from inundated territories are lower than the survival and recruitment of fledglings from territories not inundated by water [17]. Our individual-based model estimates that under current reservoir operations rising water levels that extend into yellow warbler habitat (the riparian shrub zone at > 435m a.s.l.) reduce annual productivity by 37%, from an average of 1.62 independent young per female to 1.06 independent young per female (Fig 2, scenarios 4 and 2). Our model predicts that annual productivity would be improved, to an average of 1.44 independent young per female, by delaying when water is stored in Arrow Lakes Reservoir such that water levels do not reach 435m a.s.l until after June 12 (Fig 2, scenario 3). For context, if annual adult survival was 0.64 and juvenile survival was 0.48, annual productivity would need to average 1.5 independent young per females for a population to be self-sustaining. The standardized effect of changing reservoir operations from scenario 2 to scenario 3 or 4 on the estimated number of independent young produced is dependent on the degree to which inundation reduces post-fledging survival. However, the standardized effects remain > 0.5 and > 1.0, meaning that the differences in productivity are greater than one-half and one standard deviation, even when estimates of post-fledging survival on inundated territories are doubled. Our empirical and modeling work therefore highlights the importance of considering the post-fledging period when assessing the impact of anthropogenic activity and mitigation measures on breeding birds.

Yellow warbler productivity whether measured in the field or estimated using our individual-based model was highly variable from one year to the next. Field crews monitoring all nesting attempts made by approximately 35 females over 12 seasons found that the average number of young fledged per female ranged from 1.37 to 2.75. Our model simulating the reproduction of 35 females under the same management scenario over 50 years produced annual estimates of productivity that ranged from 1.11 to 2.86. The marked variation can be partially attributed to annual and individual variation in the timing of breeding that has knock-on consequences for reproductive success due to seasonal effects on the probability of re-nesting after the failure or success of an earlier nesting attempt. Annual variation in the timing of breeding in yellow warblers is associated with conditions on migration; males arrive and females lay earlier in years where cross winds are relatively weak [30]. Individual variation is associated with age [25, 30] and whether females have prior experience of breeding in the area (Pavlik, *pers. comm.*). Stochastic events and processes, including the presence of a breeding pair of American Crows *Corvus brachyrhynchos* also contribute to annual variation in productivity [26]. The two-fold variation in productivity observed in this study highlight the difficulty of assessing reservoir impacts on birds by monitoring annual variation in breeding performance in the field.

Individual-based models that can simulate decades of breeding are able to assess the likely impact of alternative management scenarios [18]. However, these models also illustrate the role of environmental and demographic stochasticity on the productivity of small populations [20, 21]. Here, differences in the estimated productivity of yellow warblers under Scenario 1 and 3 are negligible even though the reservoir water levels during the breeding season in Scenario 3 includes the sub-set of years where reservoir impacts are expected to be reduced. Random selection of years from the entire 50-year dataset and the reduced 28-year dataset and random selection of the timing of breeding, that are independent as winter snowfall and spring precipitation influence reservoir water levels [31] while wind speed during spring migration influences the timing of breeding [30], combine to mask water level management effects on productivity when estimated over a 50-year period. Models that are run for 1000 years produce average estimates that are consistent across runs and that differ predictably across scenarios but use a timeframe that is less relevant to managers. Despite the variance associated with

environmental and demographic stochasticity the models used in this study nevertheless provide evidence that delaying reservoir fill by 14 days will significantly increase the number of independent young produced (Table 2; Scenario 2 to 3).

Productivity estimates for yellow warblers in Arrow Lakes Reservoir were sensitive to changes in daily nest survival rate, but not rates of cowbird brood parasitism rate and nest abandonment following parasitism. The limited population-level impact of brood parasitism by cowbirds is surprising given that brood parasitism is often associated with the removal of a yellow warbler egg, and parasitized nests when successful produce, on average 1.4 fewer fledglings [25]. The discrepancy between the impact of brood parasitism on individuals and the estimated impact on population-level productivity may be explained by the low daily nest survival of yellow warbler nests, which means that more than half of all nests initiated fail, and females readily re-nest after nest failure. The population level impact of cowbirds would be greater if cowbirds attack and "farm" nests of their hosts [33], brood parasitism reduces daily nest survival (e.g. [34]), and cowbird chicks reduce host nestling condition and post-fledging survival (e.g. [35]).

Active management of reservoir water levels is likely to play an increasingly important role in the conservation of wildlife dependent on riparian habitat as hydropower continues to be the leading source of renewable energy across the world [36] and remnant riparian habitat is vulnerable to diverse anthropogenic pressures [37]. Here we show, for one species, that delaying when rising water levels begin to inundate riparian shrub habitat would significantly increase annual productivity and the likelihood that the population is self-sustaining. Impacts on other bird species will differ depending on where birds nest, when birds breed relative to the timing of water level rises, and how individuals respond to flooding of their habitat. For example, willow flycatcher *Empidomax trailii* nest at lower heights and initiate breeding later than yellow warblers making their nests more vulnerable to flooding [16]. Savannah sparrrows *Passer sandwichensis*, that nest on the ground, disperse in response to rising water levels and their productivity will depend on the quality of the alternative habitat [20]. Quantifying the impacts of reservoir operations for multiple species could be accomplished using individual-based models simplified depending on the data available and parameterized using species specific data on the timing of breeding, nesting behaviour, and demography from the literature. This undertaking, although not trivial, would allow the environmental impacts of dams and reservoir operations on breeding birds to be fully incorporated into water management decisions.

## Supporting information

**S1 Table. Model estimated productivity ± SD of female yellow warblers (average number of fledglings per female and average number of independent young per female) over 50 breeding seasons under four reservoir water level management scenarios.** Productivity estimates are presented using the baseline parameters (see Table 1 for source of parameter estimate) and after individual parameters are modified by fixed amounts. Scenario 1 = long term conditions (ie the reservoir water levels observed between 1968 and 2017; Scenario 2 = early fill years (when water levels reach 435m a.s.l. by June 12); Scenario 3 = late fill years (when water levels do not reach 435m a.s.l. until after June 12); Scenario 4 = zero flooding (where reservoir water levels never exceed 435 m a.s.l).
(DOCX)

**S1 Data. Matlab files.**
(DOCX)

## Acknowledgments

We thank Ed Hill for his commitment to this long-term project, and John Cooper, Suzanne Beauchesne and Harry van Oort for their ongoing support. We are indebted to Sam Quinlan, Christine Rock, Michaela Martin, Lena Ware, Michal Pavlik and the many field technicians for access to their data and dedication in the field.

## Author Contributions

**Conceptualization:** Mathew Hepp, David J. Green.

**Data curation:** Mathew Hepp, David J. Green.

**Formal analysis:** Mathew Hepp, Eirikur Palsson, Sarah K. Thomsen, David J. Green.

**Funding acquisition:** David J. Green.

**Investigation:** Mathew Hepp, David J. Green.

**Methodology:** Mathew Hepp, Eirikur Palsson, Sarah K. Thomsen, David J. Green.

**Project administration:** David J. Green.

**Resources:** David J. Green.

**Software:** Eirikur Palsson.

**Supervision:** David J. Green.

**Validation:** Eirikur Palsson, David J. Green.

**Visualization:** Sarah K. Thomsen.

**Writing – original draft:** Mathew Hepp, Sarah K. Thomsen.

**Writing – review & editing:** Eirikur Palsson, Sarah K. Thomsen, David J. Green.

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
