## [Decision Letter · Decision Letter 0]

13 Oct 2020

PONE-D-20-17480

Predicting the effects of reservoir water level management on the reproductive output of a riparian songbird

PLOS ONE

Dear Dr. Green,

Thank you for submitting your manuscript to PLOS ONE. After careful consideration, we feel that it has merit but does not fully meet PLOS ONE’s publication criteria as it currently stands. Therefore, we invite you to submit a revised version of the manuscript that addresses the points raised during the review process.

Thank you for your patience as I  worked diligently to find reviewers for this work. Ultimately, one reviewer read the paper and found it interesting and potentially sound, but lacking some methodological details needed to fully vet it. I agree with these comments, and also with the suggestion to broaden the scope of the paper beyond implications for this one study site. I look forward to the revised submission.

We look forward to receiving your revised manuscript.

Kind regards,

Matthew Shawkey

Academic Editor

PLOS ONE

Journal Requirements:

2. In your Methods section, please provide additional location information of the study site, including geographic coordinates for the data set if available.

"Funding for this work came from National Science and Research Council Discovery grants to DJG (RGPIN 2009-261899, 2014-05798, 2019-05513) and EP (RGPIN-2016-04625) and a Columbia Basic Fish and Wildlife Compensation Program (http://fwcp.ca) grant to DJG (SPI 4639).  Fieldwork was also indirectly supported through a BC Hydro Water License Requirements contract to Cooper Beauchesne and Associates Ltd (CLBMON-36). The funders had no role in study design, data collection and analysis, decision to publish, or preparation of the manuscript."

We note that you received funding from a commercial source: Cooper Beauchesne and Associates Ltd.

Reviewers' comments:

Reviewer's Responses to Questions

**Comments to the Author**

1. Is the manuscript technically sound, and do the data support the conclusions?

Reviewer #1: Partly

2. Has the statistical analysis been performed appropriately and rigorously? 

Reviewer #1: No

3. Have the authors made all data underlying the findings in their manuscript fully available?

Reviewer #1: No

4. Is the manuscript presented in an intelligible fashion and written in standard English?

Reviewer #1: Yes

5. Review Comments to the Author

Reviewer #1: I have reviewed the paper by Hepp et al. on predicting how flooding regimes affects yellow warbler productivity using an individual-based model. The paper is clearly written, although there were several methodological details that were missing for a complete understanding of the model and its results. I found the introduction to be informative and described previous literature, but the discussion could be broadened to discuss the implications of this research beyond the study site. More detailed comments are below.

Abstract:

- Third paragraph: (0.56 +/- xx)

METHODS:

In the section: “Breeding biology, daily nest survival rates and re-nesting probability”

“We evaluated whether clutch size…” - what were the models that you created here? Poisson regression for clutch size? Logistic regression for re-nesting? More detail is needed.

Why are you selecting the reservoir level randomly? Wouldn’t you want to have this be a deterministic variable? (Or is it sampling without replacement? Okay, yes it is - why random in that case? And how do you get to 50 years for Scenarios 2 and 3 if there are fewer than 50 years in the dataset?)

“Date, ground elevation and nest height are randomly chosen from a normal distribution, where the mean and SD are calculated from the monitoring data.” - please provide more information about these data and their collection. Did you collect these data from all monitored nests? This is an important variable for the model, and so it would be good to have more information on it. Is it possible that females altered their nest heights in response to reservoir conditions? In which case, it would make more sense to draw from a year-specific distribution of nest heights/elevations instead of a global distribution.

Where are the nest elevation data coming from?

Table 1: What is DOY?

Model validation and sensitivity:

- Why are you only validating Scenario 2? If anything, wouldn’t you want to validate Scenario 1, which represents the historical conditions? I’m also confused how this is really a validation. The output of the IBM is a direct consequence of the data put into it, there are no emergent properties. A validation would indicate that you withheld some data and are predicting it, but if so, then you need to describe this process.

- Where did this radiotelemetry data come from? Is this a separate dataset that you are using for validation?

- I’m confused by the equation of this section. Why are you comparing 1or2 and 3or4?

- Section needs more explanation for why sensitivity levels were chosen (e.g., 50% or 1%).

RESULTS:

- You need to provide methods for all tests (e.g., no methods described comparing ground elevation and date, and other F-stats in first paragraph).

- If so few nests failed from flooding (22/522 or 4%), why is flooding the main source of concern for nest failure?

Figure 2: It’s surprising to me that Scenario 3 would have lower median productivity than Scenario 1, seeing as Scenario 3 is just the “good years” from Scenario 1. Can you explain?

Also, what are the box plots showing (what are the box boundaries and “whiskers” indicating)?

Table 2: It is difficult to read this table due to formatting issues, not entirely clear which numbers go with which row.

Why is there a difference in the effect for number of fledglings vs. independent young? Do fledglings die from flooding? If so, this needs to be stated more clearly in the methods.

I appreciate the use of standardized coefficients, but I think it would be more useful to put the effects in terms of number of birds produced - a more intuitive difference in the scenario. You do this in the discussion, but I think those numbers should be put in the results.

DISCUSSION:

You discuss large differences in productivity among the scenarios, but Figure 2 indicates a lot of uncertainty around these estimates that makes it seem like they may not be all that different. Please discuss this uncertainty and what it means for your conclusions.

6. PLOS authors have the option to publish the peer review history of their article (what does this mean?). If published, this will include your full peer review and any attached files.

Reviewer #1: No

---

## [Author Response · Author response to Decision Letter 0]

14 Jan 2021

 I believe we have revised the manuscript to meet the style requirements outlined in the templates.

2. In your Methods section, please provide additional location information of the study site, including geographic coordinates for the data set if available.

 We have added the geographic coordinates of the study area

"Funding for this work came from National Science and Research Council Discovery grants to DJG (RGPIN 2009-261899, 2014-05798, 2019-05513) and EP (RGPIN-2016-04625) and a Columbia Basic Fish and Wildlife Compensation Program (http://fwcp.ca) grant to DJG (SPI 4639). Fieldwork was also indirectly supported through a BC Hydro Water License Requirements contract to Cooper Beauchesne and Associates Ltd (CLBMON-36). The funders had no role in study design, data collection and analysis, decision to publish, or preparation of the manuscript."

We note that you received funding from a commercial source: Cooper Beauchesne and Associates Ltd.

Amended statement

Cooper and Beauchesne and Associates Ltd provided field accommodation and a food allowance for a graduate student and field assistant that collected data used in this paper in each year from 2008-2017. This does not alter our adherence to PLOS ONE policies on sharing data and materials 

Reviewers' comments:

Reviewer's Responses to Questions

Q 1-2 – we address the reviewer’s response to these questions in the comments to author

Q 3. Data availability. We are happy to post all underlying data used to obtain parameters used in the model in the Dryad data depository. We have prepared an excel file with these data.

5. Review Comments to the Author

Reviewer #1: I have reviewed the paper by Hepp et al. on predicting how flooding regimes affects yellow warbler productivity using an individual-based model. The paper is clearly written, although there were several methodological details that were missing for a complete understanding of the model and its results. I found the introduction to be informative and described previous literature, but the discussion could be broadened to discuss the implications of this research beyond the study site. More detailed comments are below.

We have broadened the discussion adding an additional paragraph describing variance arsing due to the stochastic processes included in individual based models and modifying the final paragraph to make the case that the modelling approach used here will be increasingly relevant to the management of riparian habitat associated with reservoirs worldwide.

Abstract:

- Third paragraph: (0.56 +/- xx) 

SD added

METHODS:

In the section: “Breeding biology, daily nest survival rates and re-nesting probability”

“We evaluated whether clutch size…” - what were the models that you created here? Poisson regression for clutch size? Logistic regression for re-nesting? More detail is needed.

We agree and have added information on the type of regression model that was use 

Why are you selecting the reservoir level randomly? Wouldn’t you want to have this be a deterministic variable? (Or is it sampling without replacement? Okay, yes it is - why random in that case? And how do you get to 50 years for Scenarios 2 and 3 if there are fewer than 50 years in the dataset?)

The confusion comes from an error in the text on our part. The years were sampled with replacement so are randomly drawn from those available for a particular scenario. We have corrected our mistake.

“Date, ground elevation and nest height are randomly chosen from a normal distribution, where the mean and SD are calculated from the monitoring data.” - please provide more information about these data and their collection. Did you collect these data from all monitored nests? This is an important variable for the model, and so it would be good to have more information on it. Is it possible that females altered their nest heights in response to reservoir conditions? In which case, it would make more sense to draw from a year-specific distribution of nest heights/elevations instead of a global distribution.

We did collect data on the ground elevation and nest heights for all nests and describe the data collection in the Breeding biology section. We show in the results that the nest heights do not change with date as would be expected if females respond to reservoir conditions where water levels rise as the season progresses. Females also do not change territories or move to higher elevations within a territory as the season progresses, establishing territories prior to any change in the reservoir level. Since females do not alter their nest heights in response to reservoir conditions we feel that the use of a global distribution makes sense. 

Where are the nest elevation data coming from?

The nest elevation data is the sum of the ground elevation and nest height and defined in the text describing the re-nesting model

Table 1: What is DOY?

DOY is used as an abbreviation for Day of Year, where January 1 = 1. We have added a footnote to the Table so that the abbreviation is defined

Model validation and sensitivity:

- Why are you only validating Scenario 2? If anything, wouldn’t you want to validate Scenario 1, which represents the historical conditions? I’m also confused how this is really a validation. The output of the IBM is a direct consequence of the data put into it, there are no emergent properties. A validation would indicate that you withheld some data and are predicting it, but if so, then you need to describe this process.

We validate the model output against Scenario 2 as the years when the fieldwork was conducted almost all fall in this scenario (11 of the 12 years of fieldwork). We state this in the section describing the scenarios but have now clarified why we compare the field data with the model output from this scenario. 

We agree that there are no emergent properties, but use the comparison of the IBM and the observational data to show that the processes we have included in the model captures reproduction and breeding season for this species, suggesting that we have not missed a key component in the breeding cycle. 

- Where did this radiotelemetry data come from? Is this a separate dataset that you are using for validation?

The radiotelemetry data comes from reference [17]. We have added a statement that describes where this data is published

- I’m confused by the equation of this section. Why are you comparing 1or2 and 3or4?

We have re-written this section to clarify we compare scenarios 1 and 2, the long-term historical and contemporary conditions with alternatives that would be expected to reduce the negative effects of reservoirs.

- Section needs more explanation for why sensitivity levels were chosen (e.g., 50% or 1%).

We have expanded this section to provide our rationale for the levels selected.

RESULTS:

- You need to provide methods for all tests (e.g., no methods described comparing ground elevation and date, and other F-stats in first paragraph).

We have added this information to the methods

- If so few nests failed from flooding (22/522 or 4%), why is flooding the main source of concern for nest failure?

The perception prior to the study was that flooding would be far more prevalent, although this modelling exercise illustrates the importance of the post-fledging period for this species. Flooding will clearly be more important for ground-nesting species

Figure 2: It’s surprising to me that Scenario 3 would have lower median productivity than Scenario 1, seeing as Scenario 3 is just the “good years” from Scenario 1. Can you explain?

Also, what are the box plots showing (what are the box boundaries and “whiskers” indicating)?

This is a good point and one that we had missed. The results give the outcome of fifty years of breeding under different scenarios and incorporate stochasticity across years within each scenario, leading to high variance in productivity. Scenario 3 would be expected to have higher median productivity than Scenario 1 if each year was used once, but we used a repeated random draw under each scenario (see above) so when run over 50 years there was little difference in the predicted productivity in the two scenarios.

We re-ran the models for 1000 years and multiple runs of this length converge on a mean/median productivity where Scenario 3 outperforms Scenario 2 as the reviewer expected. We have not replaced the results based on the shorter time period as expectations based on the earlier analysis better capture year-to-year stochasticity in reservoir water operations and the likely outcome of any mitigation efforts. Instead we discuss the considerable variation in the productivity estimates.

The box boundaries are the 25, 75% quartiles and the whiskers extend to the largest/smallest value no further than 1.5 times the inter-quartile range. We have added this to the figure legend. 

Table 2: It is difficult to read this table due to formatting issues, not entirely clear which numbers go with which row.

We have reformatted the table

Why is there a difference in the effect for number of fledglings vs. independent young? Do fledglings die from flooding? If so, this needs to be stated more clearly in the methods.

We have made sure this is clear in the methods

I appreciate the use of standardized coefficients, but I think it would be more useful to put the effects in terms of number of birds produced - a more intuitive difference in the scenario. You do this in the discussion, but I think those numbers should be put in the results.

We have modified the table to include both 

DISCUSSION:

You discuss large differences in productivity among the scenarios, but Figure 2 indicates a lot of uncertainty around these estimates that makes it seem like they may not be all that different. Please discuss this uncertainty and what it means for your conclusions.

We have added a paragraph to discuss the variance

 We have used PACE to check our figures

---

## [Editor Report · Decision Letter 1]

5 Feb 2021

Predicting the effects of reservoir water level management on the reproductive output of a riparian songbird

PONE-D-20-17480R1

Dear Dr. Green,

We’re pleased to inform you that your manuscript has been judged scientifically suitable for publication and will be formally accepted for publication once it meets all outstanding technical requirements.

Kind regards,

Matthew Shawkey

Academic Editor

PLOS ONE

Additional Editor Comments (optional):

Thanks for your patience with the review process. I have reviewed the paper and replies, and do not think it is necessary to send it back out for review. The comments from the reviewer were helpful, and you addressed them well. Congratulations on the interesting paper.

Please make sure that the data used for parameter estimation is deposited in Dryad as soon as possible, and definitely before publication. 
---

## [Editor Report · Acceptance letter]

11 Feb 2021

PONE-D-20-17480R1 

Predicting the effects of reservoir water level management on the reproductive output of a riparian songbird 

Dear Dr. Green:

I'm pleased to inform you that your manuscript has been deemed suitable for publication in PLOS ONE. Congratulations! Your manuscript is now with our production department. 

Kind regards, 

on behalf of

Dr. Matthew Shawkey 

Academic Editor

PLOS ONE